# Experimental study on factors influencing low identification rates and spectral quality of *Streptococcus pneumoniae* using the Sepsityper kit

Hyunseul Choi,[1] Minhee Kang,[1] You-Keun Ko,[2] Hui-Jin Yu,[3] Tae Yeul Kim,[2] Hee Jae Huh,[2] Nam Yong Lee,[2] Hyun-Seung Lee[4]

**ABSTRACT**   The MBT Sepsityper Kit (Sepsityper) has a low identification rate for *Streptococcus pneumoniae*, and the underlying causes remain unclear. To elucidate these factors, we investigated five factors under controlled conditions simulating blood-stream infections: blood culture bottle type, blood culture system, matrix-assisted laser desorption/ionization time-of-flight mass spectrometry (MALDI-TOF MS) platform, lysis buffer volume used in Sepsityper, and sample preparation method used for MALDI-TOF MS analysis. The results showed that anaerobic bottles had a higher identification rate than aerobic bottles (5.6% vs 0%); however, no significant differences were observed for other factors. Replacement of the lysis and washing buffers with phosphate-buffered saline (PBS) in the Sepsityper was also performed. Replacing the lysis buffer with PBS improved the identification rates compared to unmodified protocols or replacing only the washing buffer. Groups without the lysis buffer exhibited typical gram-positive morphology on Gram staining, whereas those with the lysis buffer showed predominantly swollen gram-negative morphology. Further evaluations of blood culture bottle types (aerobic vs anaerobic), blood culture systems (BD BACTEC FX vs BACT/ALERT VIRTUO), MALDI-TOF MS platforms (MALDI Biotyper sirius vs VITEK MS), lysis buffer volumes used in Sepsityper (standard vs half volume), and sample preparation methods for MALDI-TOF MS analysis (direct transfer, extended direct transfer, and full extraction) revealed no significant differences in identification rates. These findings suggest that the lysis buffer plays a critical role in the poor identification rate of *S. pneumoniae* using Sepsityper; however, halving its volume does not significantly improve the outcomes.

**IMPORTANCE** This study addresses a critical issue in clinical microbiology: the low identification rate of *Streptococcus pneumoniae* using the widely used Sepsityper for matrix-assisted laser desorption/ionization time-of-flight mass spectrometry. Accurate and timely identification of *S. pneumoniae* is essential for diagnosing invasive pneumococcal infections, which are associated with high morbidity and mortality rates. Our findings revealed that the lysis buffer in the Sepsityper significantly damaged the bacterial cell wall, leading to low identification rates for *S. pneumoniae* specifically. Replacing the lysis and washing buffers with a phosphate-buffered saline solution markedly improved the identification accuracy, offering a potential workaround. This study highlights the need for further refinement of diagnostic protocols for *S. pneumoniae* and underscores the importance of alternative enrichment methods to improve clinical outcomes. These findings have significant implications for laboratory and patient care worldwide.

**KEYWORDS**   *Streptococcus pneumoniae*, Sepsityper, MALDI-TOF, lysis buffer

Address correspondence to Tae Yeul Kim, taeyeul.kim@samsung.com, or Hyun-Seung Lee, ctmasaru@gmail.com.

Hyunseul Choi, Minhee Kang, and You-Keun Ko contributed equally to this article. The authors' order was determined randomly.

The authors declare no conflict of interest.

See the funding table on p. 10.

*Streptococcus pneumoniae*, or pneumococcus, is a gram-positive, lancet-shaped, facultatively anaerobic bacterium that typically occurs in pairs or short chains, with over 90 distinct serotypes (1, 2). It is a major pathogen responsible for severe infections, including pneumonia, meningitis, and bloodstream infections (BSIs) (1–4). Pneumococcal BSI is a significant global health issue, contributing to substantial morbidity and mortality worldwide (5, 6). Consequently, the rapid and accurate diagnosis of pneumococcal BSI has become a critical focus for healthcare providers and clinical microbiologists.

In modern clinical microbiology laboratories, matrix-assisted laser desorption/ionization time-of-flight mass spectrometry (MALDI-TOF MS) is the primary tool for bacterial identification (7). This method requires a sufficient bacterial biomass, typically around $1 \times 10^5$ CFU per spot (8–10). Currently, the standard method for achieving this is subculturing on solid agar plates for 18–24 h (11, 12), which is a significant bottleneck that delays the timely de-escalation of broad-spectrum antibiotics to targeted therapies. To address these delays, various pretreatment protocols for directly purifying and/or enriching bacterial pellets from positive blood culture broths have been developed. These include filtration-based (13–16), affinity capture-based (17, 18), copolymer-based (19, 20), red blood cell (RBC) lysis, centrifugation-based approaches (21–25), and short-term subculture protocols (26–28).

The MBT Sepsityper Kit (Bruker Daltonics, Bremen, Germany) is the most widely used commercial kit for pretreating positive blood culture broth to enable rapid bacterial identification using MALDI-TOF MS (13, 19). The Standard Sepsityper (SS) protocol begins with RBC lysis, followed by centrifugation and washing steps to obtain a bacterial pellet. The pellet was subjected to protein extraction using ethanol, formic acid, and acetonitrile. According to a meta-analysis of 21 studies, the SS protocol achieved correct species-level identification in 79.8% of positive blood culture bottles. However, the identification rates for gram-positive bacteria (76.1%) and yeasts (65.9%) are much lower than those for gram-negative bacteria (89.6%) (29). To improve efficiency, Bruker Daltonics recently introduced the Rapid Sepsityper (RS) protocol, which eliminates the protein extraction step. This protocol reduces hands-on time from 20 to 30 min to less than 10 min and enables identification results within 15–20 min of a positive blood culture alert.

The MBT Sepsityper module (Bruker Daltonics), a software module optimized for analyzing mass spectra obtained using the Sepsityper, has significantly improved the species-level identification rates for gram-positive bacteria and yeasts (30, 31). However, despite these advancements, the correct species-level identification rate for *S. pneumoniae* remains extremely low (less than 20%) (30, 32). Furthermore, the Sepsityper has shown poorer performance for *S. pneumoniae* than in-house centrifugation-based methods (33–35). Given the diagnostic importance of pneumococcal BSI and the widespread use of the Sepsityper, addressing the low identification rate of *S. pneumoniae* is a critical issue. Hypotheses to explain this issue include the susceptibility of *S. pneumoniae* to autolysis (36, 37), its vulnerability to the lysis buffer (32, 38), and its interaction with blood culture media components (14, 39–41). Other factors, such as blood culture bottle type (42, 43), database quality (44, 45), and extraction methods (39, 44, 46, 47), may also affect the spectral quality and identification rates during MALDI-TOF MS analysis. Despite these insights, the specific reasons for the low identification rate of *S. pneumoniae* using the Sepsityper remain unclear, and a definitive solution has yet to be found.

This study aimed to investigate the underlying causes of the low identification rate of *S. pneumoniae* and evaluate five key factors influencing its identification using the Sepsityper under well-controlled experimental conditions simulating BSI: blood culture bottle type, blood culture system, MALDI-TOF MS platform, lysis buffer volume used in the Sepsityper, and sample preparation method used for MALDI-TOF MS.

## MATERIALS AND METHODS

### Blood cultures

*S. pneumoniae* ATCC 49619 and five clinical strains of *S. pneumoniae* isolated from patients with BSI were cultured at 37°C with 5% $CO_2$ on blood agar plates (BAP) for 18 h. The characteristics of these strains are shown in Table S1. Following incubation, 3–4 colonies were suspended in 0.9 × phosphate-buffered saline (PBS) to a 0.5 McFarland standard. This suspension was serially diluted to achieve a concentration of 10–100 CFU/mL. Next, 0.5 mL of the diluted suspension was mixed with 10 mL of whole blood from healthy volunteers. The mixture was introduced into aerobic and anaerobic bottles and incubated in their respective blood culture systems: BACT/ALERT FA PLUS and FN PLUS bottles in the BACT/ALERT VIRTUO system (bioMérieux, Marcy l'Etoile, France) and BD BACTEC Plus Aerobic/F and Lytic/10 Anaerobic/F bottles in the BD BACTEC FX system (BD, Franklin Lakes, NJ, USA). Once a positive alarm was triggered, the bottles were removed and processed using the Sepsityper.

### Pellet acquisition procedure

Pellets were obtained using the Sepsityper using two methods: the standard method, as per the manufacturer's instructions, and the Cordovana method, which uses 100 µL of lysis buffer, half the volume used in the standard method (32). For the standard method, positive blood culture broth (1 mL) was transferred to a 1.5 mL microcentrifuge tube. After adding 200 µL of lysis buffer, the sample was vortexed for 10 s and centrifuged at 15,000 × *g* for 2 min. The supernatant was discarded, and the pellet was resuspended in 1 mL washing buffer, followed by centrifugation at 15,000 × *g* for 1 min. The final pellet was collected.

For the Cordovana method (32), the protocol was modified by using half the standard volume of lysis buffer (100 µL). After centrifugation, a double-layer pellet was formed, with RBCs at the bottom and bacteria on top. The bacterial layer was carefully transferred to a new microcentrifuge tube, resuspended in 1 mL washing buffer, and centrifuged again at 15,000 × *g* for 1 min. The pellets were collected. If RBCs were still present in the pellet, an additional washing step was performed.

### Sample preparation procedures

Three protocols recommended by the manufacturer were used: the RS protocol with the direct transfer (DT) method, the RS protocol with the extended direct transfer (eDT) method, and the SS protocol with the full extraction (Ext) method. The pellet was spotted at two positions on a MALDI target plate using a toothpick. The first spot did not receive formic acid treatment (DT method). The second spot was overlaid with 1 µL of 70% formic acid and dried at room temperature (eDT method). For the full extraction method (Ext method), the pellet was resuspended in 300 µL of distilled water and 900 µL of ethanol and centrifuged at 15,000 × *g* for 2 min. The supernatant was discarded, and the sample was centrifuged again at 15,000 × *g* for 2 min. After removing the supernatant, the pellet was air-dried at room temperature for 5 min. Depending on the pellet size, 20–50 µL of 70% formic acid and an equal volume of acetonitrile were added, followed by centrifugation at 15,000 × *g* for 2 min. Finally, 1 µL of the supernatant was spotted onto a MALDI target plate and dried at room temperature. Each spot was then overlaid with 1 µL of matrix solution, before MALDI-TOF MS analysis.

### MALDI-TOF MS analysis

Mass spectra were acquired and processed using two MALDI-TOF MS platforms: the MALDI Biotyper sirius (Bruker Daltonics), along with the MBT Compass HT IVD Library v2023, and the VITEK MS (bioMérieux, Marcy l'Etoile, France), along with the VITEK MS v3.2 Knowledge Base. The MALDI Biotyper sirius was operated using the MBT Compass HT IVD software (v5.2.310), incorporating the MBT Sepsityper module. Species-level

identification was considered reliable for score values of ≥1.6. Similarly, for VITEK MS, species-level identification was deemed reliable for confidence values ≥60%.

## Scanning electron microscopy analysis

To evaluate the impact of RBCs and broth media on *S. pneumoniae* morphology, scanning electron microscopy (SEM) analysis was performed. A 0.5 mL suspension of *S. pneumoniae* ATCC 49619 (10–100 CFU/mL) was mixed with 10 mL of either whole blood or 0.9 × PBS, introduced into a BACT/ALERT FA PLUS bottle, and incubated in the BACT/ALERT VIRTUO system. After a positive alarm, the bottles were immediately removed, and the positive blood culture broth was collected and centrifuged at 15,000 × *g* for 1 min. The supernatant was discarded, and the pellet was washed three times with 0.9 × PBS before SEM analysis. For comparison, a suspension of *S. pneumoniae* ATCC 49619, adjusted to a 3.0 McFarland standard, was prepared and directly analyzed using SEM without blood culture. The samples were placed on glass slides, fixed with 2.5% glutaraldehyde, and their morphology was examined using field-emission SEM (SU-8010; Hitachi Ltd., Tokyo, Japan) at 4.0 kV.

## Sepsityper reagent modification study

To identify reagents interfering with *S. pneumoniae* identification, the lysis and washing buffers in a critical step in the Sepsityper were replaced with PBS. *S. pneumoniae* ATCC 49619 and five clinical strains were cultured on BAP for 18 h and suspended in 0.9 × PBS (3.0 McFarland standard). The bacterial suspension was processed using the Sepsityper, following the RS protocol with the DT method. To investigate whether the reagents were responsible for the low identification rate of *S. pneumoniae*, the lysis and washing buffers in the Sepsityper were replaced with 0.9 × PBS. MALDI-TOF MS analysis was conducted using VITEK MS, with nine replicates per sample. Gram staining of the pellet remnants was performed for comparison.

## Gram staining of a representative gram-positive cocci pellet

To evaluate the impact of lysis buffer on the bacterial cell wall of gram-positive cocci, Gram staining of the pellet was performed using several gram-positive cocci: *Streptococcus mitis* ATCC 49456, *Streptococcus oralis* ATCC 9811, *Streptococcus pyogenes* ATCC 19615, *Streptococcus agalactiae* ATCC 13813, *Staphylococcus aureus* ATCC 19636, *Staphylococcus epidermidis* ATCC 35983, *Enterococcus faecalis* ATCC 19433, and *Enterococcus faecium* ATCC 19434. Each strain was cultured on BAP for 18 h and suspended in 0.9 × PBS (3.0 McFarland standard). Pellets were obtained using the Sepsityper without reagent modification.

## Pilot study on factors influencing Sepsityper performance

In the pilot study, *S. pneumoniae* ATCC 49619 was used. Blood culture bottles were spiked with bacteria and cultured in BACT/ALERT aerobic and anaerobic bottles incubated in the BACT/ALERT VIRTUO system, and BD BACTEC aerobic and anaerobic bottles incubated in the BD BACTEC FX system. Nine replicates were prepared for each bottle type and blood culture system. All positive blood culture bottles were processed using the Sepsityper with a standard or half the standard volume of lysis buffer. Samples were prepared using the DT, eDT, or Ext methods, and MALDI-TOF MS analysis was conducted using the MALDI Biotyper sirius or VITEK MS platforms.

## Validation study on factors influencing Sepsityper performance

For the validation study, five clinical strains of *S. pneumoniae* isolated from patients with BSI were used. Blood culture bottles spiked with clinical strains were prepared, with a single bottle prepared for each strain, and processed as in the pilot study; however, MALDI-TOF MS analysis was performed exclusively using the MALDI Biotyper sirius platform in duplicate.

## Statistical analysis

The time to detection (TTD) was defined as the interval from inserting the bottle into the instrument to the positive alarm. One-way analysis of variance (ANOVA) was used to compare the TTD values across the bottle types. The $\chi^2$ test (with or without Yates' correction) was used to assess differences in correct species-level identification rates. Firth-type logistic regression analysis or generalized estimating equations were used to identify independent factors affecting species-level identification rates. Statistical significance was set at $P < 0.05$. Analyses were conducted using the Statistical Package for the Social Sciences v29.0.2.0 (IBM Corp., Armonk, NY, USA) and R v4.4.3 (Vienna, Austria; http://www.R-project.org/).

## RESULTS

SEM revealed no notable differences in bacterial cell size, shape, or arrangement between groups exposed to both broth media and RBCs, broth media alone in the blood culture bottle, and the group without blood culture. No damage to bacterial cells was observed in any group (Fig. S1).

Replacing the lysis buffer (98.2%, 53/54) or both the lysis and washing buffers (100%, 54/54) in the Sepsityper with PBS significantly improved species-level identification rates for *S. pneumoniae*. These rates were markedly higher than those observed in the group with no reagent modification (48.2%, 26/54) and the group in which only the washing buffer was replaced with PBS (42.6%, 23/54; Table 1).

The Gram staining results were consistent with these findings. Groups in which the lysis buffer or both buffers were replaced with PBS predominantly showed typical gram-positive diplococcal lancet morphology. In contrast, groups treated with lysis buffer (no modification or replacing of the washing buffer with PBS) primarily showed swollen or empty gram-negative cells, with few positive cocci (Fig. 1). Typical gram-positive cells were observed in *S. mitis*, *S. oralis*, *S. pyogenes*, *S. agalactiae*, *S. aureus*, *S. epidermidis*, *E. faecalis*, and *E. faecium* with lysis buffer (no modification; Fig. 2).

All 36 blood culture bottles spiked with *S. pneumoniae* ATCC 49619 produced a positive alarm, with a TTD ranging from 11 to 13 h. No significant differences in mean TTD were observed between the bottle types or blood culture systems. All positive blood culture bottles were processed, and MALDI-TOF MS analysis was performed according to the protocol of the pilot study. As shown in Table S2 *S. pneumoniae* was correctly identified at the species level in 12 of 432 MALDI-TOF MS analyses (2.8%). The identification rates were significantly higher ($P < 0.001$) in anaerobic bottles (12/216, 5.6%) than in aerobic bottles (0/216, 0%). However, no significant differences in identification rates were observed based on the blood culture system (BD BACTEC FX [3/216, 1.4%] vs BACT/ALERT VIRTUO [9/216, 4.2%], $P = 0.066$]), MALDI-TOF MS platform (MALDI Biotyper sirius [5/216, 2.3%] vs VITEK MS [7/216, 3.2%], $P = 0.793$]), volume of lysis buffer (standard [7/216, 3.2%] vs half volume [5/216, 3.2%], $P = 0.793$]), and sample preparation method

**TABLE 1** Results of the Sepsityper reagent modification study[a]

| Protocol | Reagent modification | | Identification rate | *P*-value |
|---|---|---|---|---|
| | **Lysis buffer** | **Washing buffer** | | |
| No reagent modification | X | X | 26/54 (48.2%) | —[b] |
| Washing buffer replaced with PBS | X | O | 23/54 (42.6%) | 0.699 |
| Lysis buffer replaced with PBS | O | X | 53/54 (98.2%) | <0.001 |
| Washing and lysis buffers replaced with PBS | O | O | 54/54 (100.0%) | <0.001 |

[a]X, no reagent modification; O, reagent modification by replacing with PBS.
[b]"—", "no *P*-value.

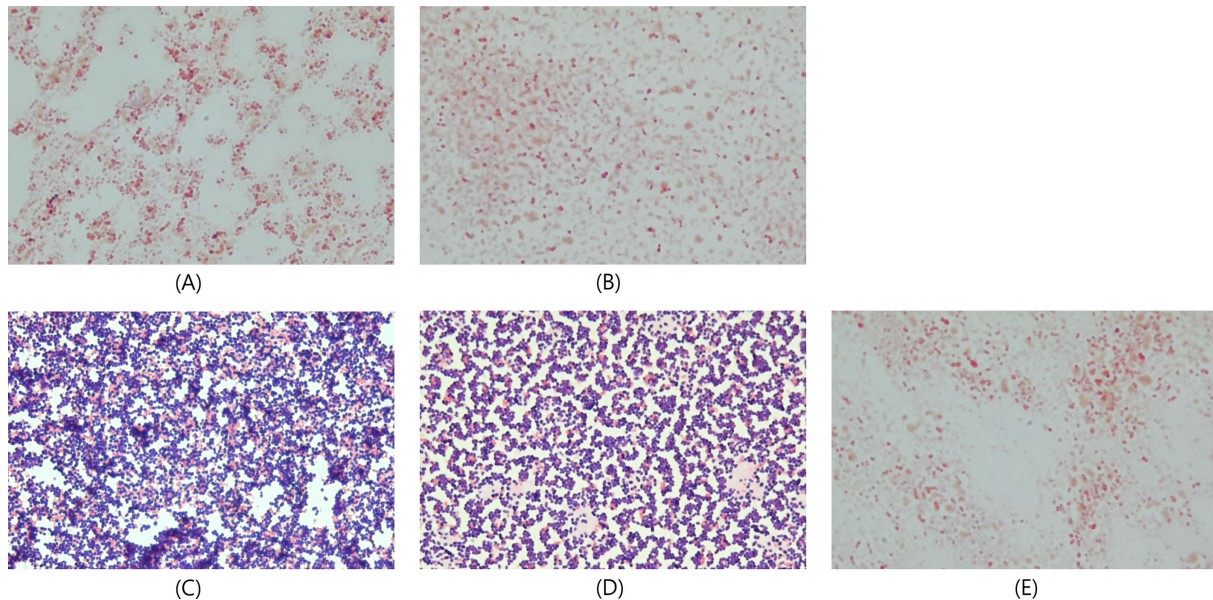

**FIG 1** Gram staining of pellets in the Sepsityper reagent modification study. Gram staining results of *S. pneumoniae* ATCC 49619 pellets obtained after 18 h of culture on BAP. Pellets were processed using the following Sepsityper reagent conditions: (A) no reagent modification, (B) washing buffer replaced with PBS, (C) both lysis and washing buffers replaced with PBS, (D) lysis buffer replaced with PBS, and (E) half volume of lysis buffer.

(DT [3/144, 2.1%] vs eDT [2/144, 1.4%] vs Ext [7/144, 4.9%, *P* = 0.234]). Firth-type logistic regression analysis revealed that the blood culture bottle type was the only independent predictor of correct species-level identification rates for *S. pneumoniae* (Table 2).

In the validation study, 16 of the 20 blood culture bottles spiked with five clinical strains of *S. pneumoniae* produced positive alarms within 16 h. The remaining four bottles—one BACT/ALERT anaerobic bottle and three BD BACTEC Plus Aerobic/F bottles—produced positive alarms after 16 h and were excluded from the analysis due to suspected technical errors. Clinical strains spiked in the excluded four bottles were as follows: serotype 3 (*n* = 2), serotype 35B (*n* = 1), and serotype 15C (*n* = 1). The correct species-level identification rate in the validation study was 7.8% (15/192). Consistent with the pilot study findings, the identification rates were significantly higher in anaerobic bottles (11/72, 15.3%) than in aerobic bottles (4/120, 3.3%, *P* = 0.007). No significant differences in identification rates were observed based on the blood culture system (BD BACTEC FX [4/84, 4.8%] vs BACT/ALERT VIRTUO [11/108, 10.2%, *P* = 0.264]), volume of lysis buffer (standard [4/96, 4.2%] vs half volume [11/96, 11.5%, *P* = 0.107]), and sample preparation method (DT [4/64, 6.3%] vs eDT [3/64, 4.7%] vs Ext [8/64, 12.5%, *P* = 0.199]; Table S3). Unlike in the pilot study, logistic regression analysis in the validation study identified both blood culture bottle type and the volume of lysis buffer as significant factors influencing the correct species-level identification rates for *S. pneumoniae* (Table

**TABLE 2** Regression analysis for factors influencing correct species-level identification rates of *S. pneumoniae* ATCC 49619[a]

| Factors | Coefficient | SE | *P*-value[b] | OR (95% CI) |
|---|---|---|---|---|
| Blood culture system | −1.056 | 0.592 | 0.081 | 0.348 (0.085, 1.132) |
| Blood culture bottle type | 3.284 | 1.316 | < 0.001 | 26.732 (3.471, 116.68) |
| Volume of lysis buffer | −0.335 | 0.540 | 0.564 | 0.716 (0.217, 2.233) |
| Extraction method (DT vs eDT) | −0.351 | 0.793 | 0.676 | 0.704 (0.114, 3.736) |
| Extraction method (DT vs Ext) | 0.825 | 0.632 | 0.204 | 2.281 (0.646, 9.754) |
| MALDI-TOF MS system | 0.335 | 0.540 | 0.564 | 1.398 (0.388, 4.631) |

[a]OR, Odds ratio; CI, confidence interval.
[b]Because some results contained zero, Firth-type logistic regression analysis with rare events was used to identify independent factors influencing the correct species-level identification rates.

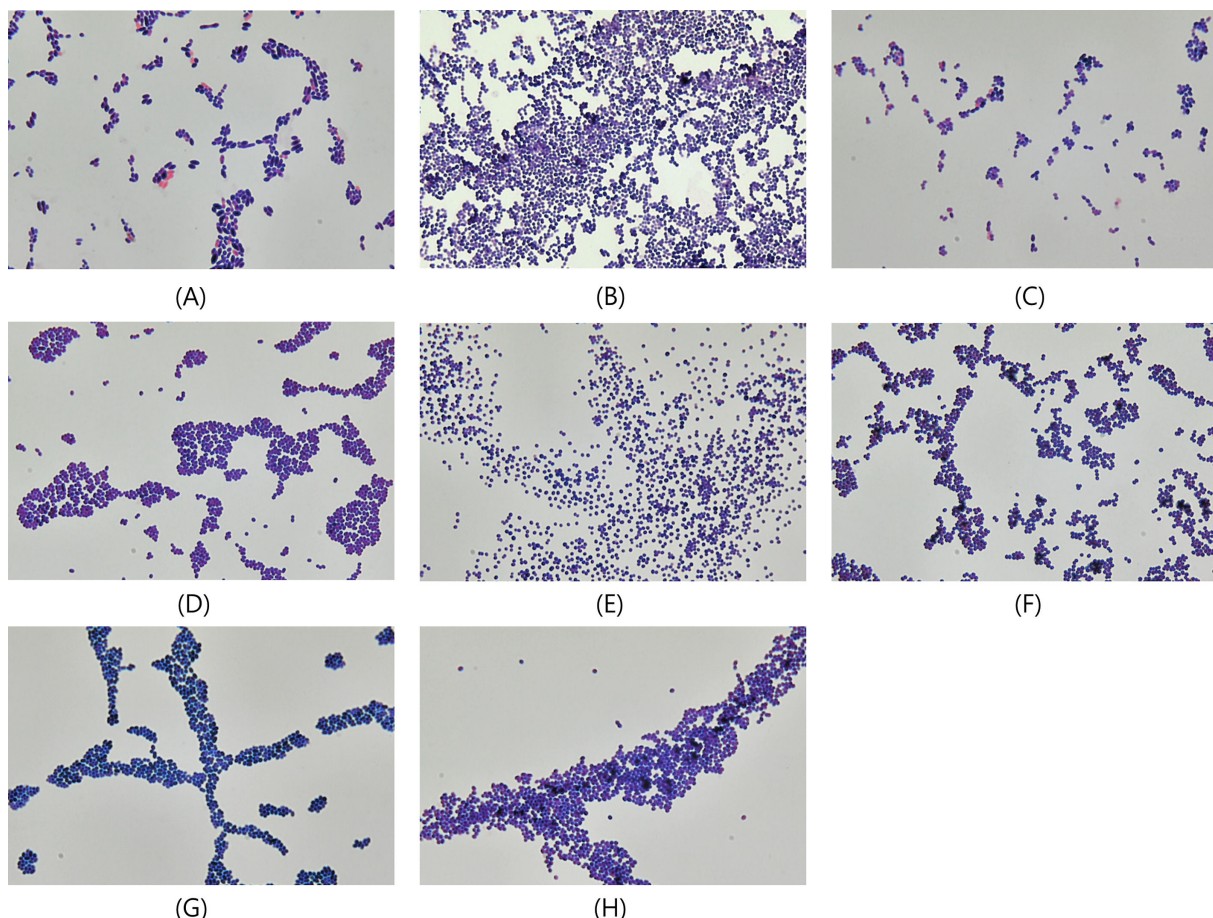

**FIG 2** Gram staining of pellets for Gram-positive cocci using the Sepsityper. Gram staining results of each ATCC strain pellet obtained after 18 h of culture on BAPs. Pellets were processed using the following gram-positive cocci: (A) *S. mitis* ATCC 49456, (B) *S. oralis* ATCC 9811, (C) *S. pyogenes* ATCC 19615, (D) *S. agalactiae* ATCC 13813, (E) *S. aureus* ATCC 19636, (F) *S. epidermidis* ATCC 35983, (G) *E. faecalis* ATCC 19433, and (H) *E. faecium* ATCC 19434.

3). Mass spectra randomly selected from the validation study were compared to a mass spectrum of *S. pneumoniae* grown on a BAP. Despite correct species-level identification, the number of mass peaks obtained using the Sepsityper—regardless of the extraction method—was markedly lower than that of the BAP-grown isolate (Fig. 3).

## DISCUSSION

This study is the first to elucidate the causes of the low identification rate of *S. pneumoniae* using the Sepsityper and to evaluate numerous factors influencing identification rates. Our results demonstrate that the lysis buffer was the primary cause of the low identification rate. Substituting the lysis buffer with PBS in the RBC lysis step dramatically

**TABLE 3** Regression analysis for factors influencing correct species-level identification rates of five clinical strains of *S. pneumoniae*[a]

| Factors | Coefficient | SE | *P*-value[b] | OR (95% CI) |
|---|---|---|---|---|
| Blood culture system | −0.377 | 0.668 | 0.573 | 0.686 (0.185, 2.542) |
| Blood culture bottle type | 1.783 | 0.577 | 0.002 | 5.949 (1.921, 18.427) |
| Volume of lysis buffer | 1.197 | 1.197 | 0.044 | 3.310 (1.035, 10.581) |
| Extraction method (DT vs eDT) | −0.329 | 0.807 | 0.684 | 0.720 (0.148, 3.501) |
| Extraction method (DT vs Ext) | 0.856 | 0.666 | 0.199 | 2.353 (0.638, 8.683) |

[a]OR, odds ratio; CI, confidence interval.
[b]Generalized estimating equation analysis with rare events was used to identify independent factors influencing correct species-level identification rates.

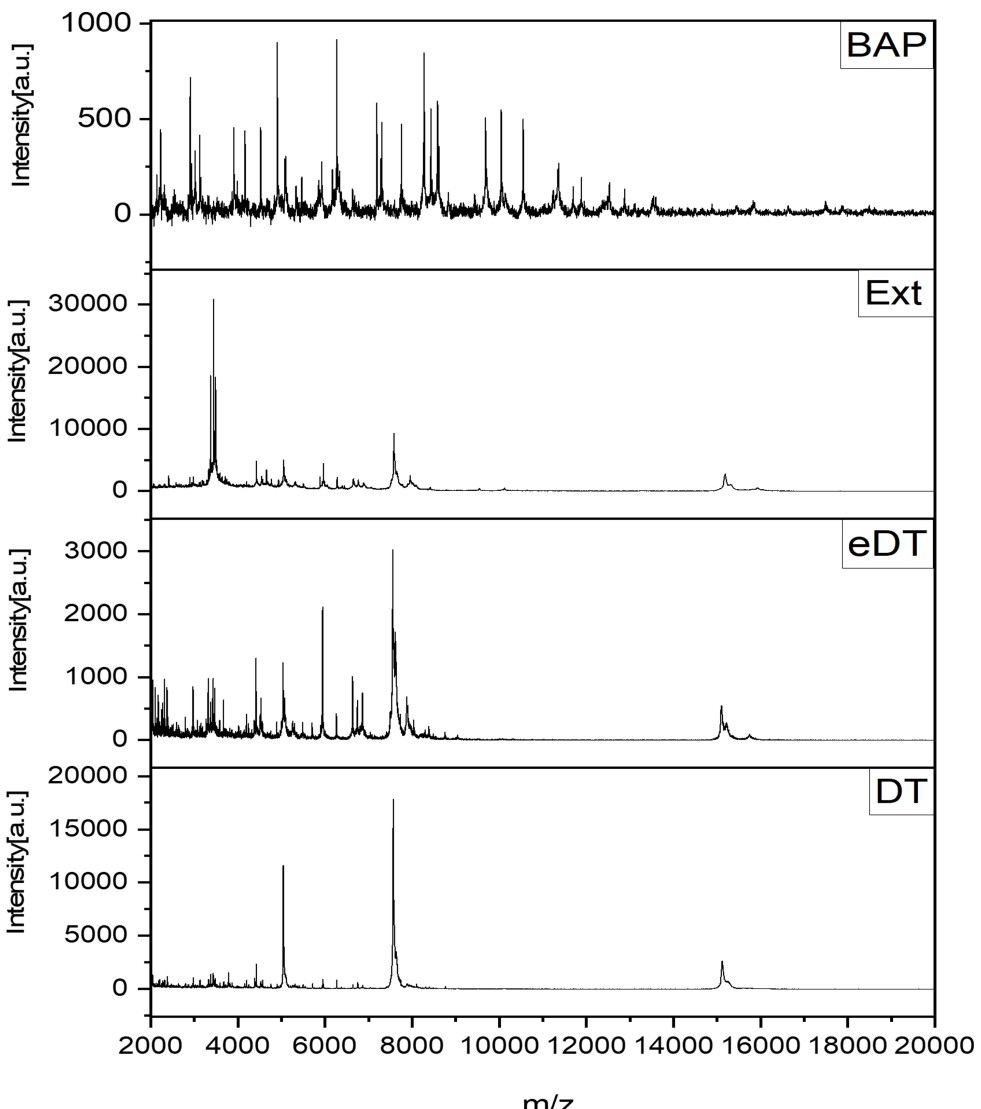

**FIG 3** A spectrum from a BAP was obtained through additional MALDI Biotyper sirius analysis of *S. pneumoniae* cultured on a BAP, while the remaining spectra were randomly selected from the validation study.

improved the identification rate, achieving near-perfect results (Table 1). Gram staining revealed that *S. pneumoniae* treated with the lysis buffer appeared gram-negative, except for a few gram-positive cocci, indicating that the lysis buffer caused considerable damage to the bacterial peptidoglycan layer (Fig. 1).

Our study reveals that the impact of lysis buffer in the Sepsityper on the bacterial cell wall may be specific for *S. pneumoniae*. Gram-positive cocci, including the closely related species of the *S. mitis* group such as *S. mitis* and *S. oralis*, which are often associated with misidentification with *S. pneumoniae* in MALDI-TOF analysis, show no considerable damage to the bacterial cell wall in the Gram staining (Fig. 2). Although the mechanism of cell wall damage by lysis buffer was not elucidated in this study, the modified bile solubility test (48) is similar because it shows a rapid cell wall degradation response specific for *S. pneumoniae* using specific substrates (desoxycholate or lysis buffer). The bile solubility test is recommended for differentiating *S. pneumoniae* from other viridans group streptococci (49), and the bile solubility is associated with the *lytA* gene, which encodes an autolysin in *S. pneumoniae* (50, 51).

We also observed a decrease in both the spectral quality and identification rates in MALDI-TOF MS analysis using the Sepsityper. Compared to the results obtained from

bacterial colonies, both the identification rates and the number of peaks obtained from Sepsityper-processed pellets were significantly reduced, regardless of the extraction method used (Table 1 and Fig. 3). These findings contrast with those of previous studies (31, 52), which reported that eDT and Ext methods significantly improved the identification rates of gram-positive bacteria when using Sepsityper. We hypothesized that the poor improvement in identification rates and spectral quality in our study was due to the loss of critical bacterial components, such as ribosomal proteins, during the RBC lysis and washing steps. This suggests that the reduced spectral quality and lower identification rates may be attributed to the quality of the obtained biomass rather than its quantity, as we obtained enough pellets in the reagent modification study.

Among the five assessed factors—blood culture bottle type, blood culture system, MALDI-TOF MS platform, lysis buffer volume, and sample preparation method—only the blood culture bottle type significantly affected the identification rates (Table 2). Anaerobic bottles were associated with significantly higher *S. pneumoniae* identification rates than aerobic bottles. This finding suggests that anaerobic bottles are more suitable for *S. pneumoniae* identification using the Sepsityper. A study (53) showed that *S. pneumoniae* in anaerobic culture had lower lytA expression, higher proliferation levels, and longer survival after 24 hours of culture than in aerobic culture. Modifying other factors, such as the MALDI-TOF MS platform, sample preparation method, or lysis buffer volume, did not significantly improve identification rates.

Our findings differ from those of Cordovana et al. (32), who reported that halving the lysis buffer volume improved the identification rates to approximately 70%. In our study, the Cordovana method did not significantly enhance identification rates, and no notable morphological differences were observed between the standard and half-volume lysis buffer groups in Gram staining (Fig. 1A and E). This discrepancy may be attributed to variations in the microbial characteristics across *S. pneumoniae* serotypes. In the current study, *S. pneumoniae* ATCC 49619 and five clinical strains were capsular serotypes (Table S1), and none were noncapsular. In the validation study using five clinical strains of *S. pneumoniae*, halving the lysis buffer volume had a notable impact on identification rates, achieving approximately three-fold higher identification rates than the standard method (standard vs half volume: 4.2%, 4/96 vs 11.5%, 11/96; $P = 0.107$; Table S3).

Our study had several limitations. Because the MALDI Biotyper sirius was provided temporarily, the MALDI-TOF MS analysis in the Sepsityper reagent modification study was conducted using the VITEK MS only. Due to a lack of laboratory resources, we tested a few capsular serotypes, including common serotypes in Korea such as 3, 10A, 19A, and 35B (54), using the MALDI Biotyper sirius only in the validation study. In the comparison of blood culture bottle type, pediatric blood culture bottles were not evaluated in this study. The additional experiments to elucidate the mechanism of cell wall damage by lysis buffer using autolysis-deficient strains/mutants for *lytA* were not performed. To overcome these limitations, further study should be required.

Given these findings, alternative approaches may be more reliable for identifying *S. pneumoniae* in positive blood culture broth (33, 35). The SST tube method, which removes RBCs without the use of a lysis buffer, achieved a 100% identification rate (8/8) (33). Similarly, saponin-based enrichment, which uses saponin as an RBC lysis solution, achieved an 80% identification rate (4/5) (35). Molecular approaches or modified bile solubility tests can also be performed directly on positive blood culture broth (48, 55). However, despite their superior performance in *S. pneumoniae* identification, these alternative methods are labor-intensive or costly, limiting their widespread adoption in routine diagnostics. Moreover, simply omitting the lysis buffer from the Sepsityper Kit may not be an effective strategy for improving *S. pneumoniae* identification rates. Hariu et al. reported that *S. pneumoniae* was not identified when the lysis buffer was omitted, in contrast to *Escherichia coli*, *Klebsiella pneumoniae*, and *Pseudomonas aeruginosa*, which were successfully identified without it (38). One should bear in mind that, despite the widespread use of rapid diagnostics, Gram staining remains an essential test for evaluating positive blood cultures, and subculturing remains the gold standard for

accurate identification—particularly for *S. pneumoniae*—albeit at the cost of additional time.

## Conclusion

The low identification rate of *S. pneumoniae* obtained using the Sepsityper is primarily caused by marked damage to the bacterial cell wall due to the lysis buffer. Although anaerobic blood culture bottles can improve identification rates, halving the lysis buffer volume did not significantly improve the results. Therefore, Gram staining should be performed if the blood culture result is positive. For the detection of *S. pneumoniae* in blood positive culture broth, alternative enrichment methods or subcultures are more appropriate for clinical use than the Sepsityper.

### ACKNOWLEDGMENTS

This study was supported by Basic Science Research Program through the National Research Foundation of Korea (NRF) funded by the Ministry of Education (RS-2022-00165990), the Future Medicine 2030 Project of Samsung Medical Center (grant number: SMX1250021), and by a grant from the Korea Health Technology R&D Project through the Korea Health Industry Development Institute (KHIDI), funded by the Ministry of Health and Welfare, Republic of Korea (grant number: RS-2024-00332244). We extend our gratitude to BD for supplying the BD BACTEC blood culture bottles and Bruker Daltonics for providing the MALDI Biotyper sirius used in this study.

### AUTHOR AFFILIATIONS

[1]Biomedical Engineering Research Center, Smart Healthcare Research Institute, Samsung Medical Center, Seoul, Republic of Korea
[2]Department of Laboratory Medicine and Genetics, Samsung Medical Center, Sungkyunkwan University School of Medicine, Seoul, Republic of Korea
[3]Department of Laboratory Medicine, Seoul Medical Center, Seoul, Republic of Korea
[4]Department of Laboratory Medicine, School of Medicine, Wonkwang University, Iksan, Republic of Korea

### AUTHOR ORCIDs

Hyunseul Choi http://orcid.org/0009-0004-1044-3945
Minhee Kang http://orcid.org/0000-0003-0330-7828
Tae Yeul Kim http://orcid.org/0000-0002-6405-5305
Hee Jae Huh http://orcid.org/0000-0001-8999-7561
Nam Yong Lee http://orcid.org/0000-0003-3688-0145
Hyun-Seung Lee http://orcid.org/0000-0003-4590-4989

### FUNDING

| Funder | Grant(s) | Author(s) |
| --- | --- | --- |
| Ministry of Education | RS-2022-00165990 | Hyun-Seung Lee |
| Samsung Medical Center, Sungkyunkwan University | SMX1250021 | Nam Yong Lee |
| Korea Health Industry Development Institute | RS-2024-00332244 | Hee Jae Huh |

### AUTHOR CONTRIBUTIONS

Hyunseul Choi, Formal analysis, Investigation, Methodology, Software, Validation, Writing – original draft, Writing – review and editing | Minhee Kang, Data curation, Formal analysis, Funding acquisition, Methodology, Project administration, Resources, Supervision, Validation, Writing – original draft, Writing – review and editing | You-Keun Ko,

Data curation, Formal analysis, Methodology, Writing – original draft, Writing – review and editing | Hui-Jin Yu, Formal analysis, Investigation, Methodology, Validation, Writing – review and editing | Tae Yeul Kim, Data curation, Formal analysis, Funding acquisition, Investigation, Methodology, Project administration, Resources, Supervision, Validation, Visualization, Writing – original draft, Writing – review and editing | Hee Jae Huh, Funding acquisition, Investigation, Project administration, Supervision, Writing – review and editing | Nam Yong Lee, Data curation, Funding acquisition, Project administration, Supervision, Writing – review and editing | Hyun-Seung Lee, Conceptualization, Data curation, Formal analysis, Investigation, Methodology, Project administration, Supervision, Validation, Visualization, Writing – original draft, Writing – review and editing

## ETHICS APPROVAL

This study was classified as non-human subject research and was exempted from ethical review by the Institutional Review Board of Samsung Medical Center.

## ADDITIONAL FILES

The following material is available online.

### Supplemental Material

**Supplemental Material (Spectrum01084-25-S0001.docx).** Tables S1 to S3; Fig. S1.

### Open Peer Review

**PEER REVIEW HISTORY (review-history.pdf).** An accounting of the reviewer comments and feedback.

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
