## [Reviewer comments · Microbiology Spectrum]

Microbiology Spectrum

Experimental study on factors influencing low identification rates and spectral quality of *Streptococcus pneumoniae* using the Sepsityper Kit

Hyunseul Choi, Minhee Kang, You-Keun Ko, Hui-Jin Yu, Tae Yeul Kim, Hee Jae Huh, Nam Yong Lee, and Hyun-Seung Lee

Corresponding Author(s): Hyun-Seung Lee, Wonkwang University College of Medicine

Review Timeline:

Submission Date:	June 2, 2025
Editorial Decision:	July 2, 2025
Revision Received:	July 9, 2025
Accepted:	July 12, 2025

Editor: Tulip Jhaveri

Reviewer(s): The reviewers have opted to remain anonymous.

Transaction Report:

DOI: <https://doi.org/10.1128/spectrum.01084-25>

Re: Spectrum01084-25 (Experimental study on factors influencing low identification rates and spectral quality of *Streptococcus pneumoniae* using the Sepsityper Kit)

Dear Dr. Hyun-Seung Lee:

Thank you for the privilege of reviewing your work. Below you will find my comments, instructions from the Spectrum editorial office, and the reviewer comments. The minor comments from Reviewer 3 have not been addressed, please address them as outlined below.

Revision Guidelines

Sincerely,
Tulip Jhaveri
Editor
Microbiology Spectrum

Reviewer #1 (Comments for the Author):

Since I was already involved in a previous review process (reviewer #3), my major comments have all been answered by the authors - thank you. Despite some minor issues, which were not considered and which are not listed in the "response to reviewers" file (?), this manuscript is suitable for publication in Microbiology Spectrum.

Please find below the minor comments as stated in my original review. Please note that the line numbers might have changed due to the newer modifications of the manuscript.

LI.31-32: Concerning the information about higher identification rates from anaerobic BC bottles in the abstract, I would like to propose to either delete this statement here or to add the information about the identification rate. At least for me, it was misleading, since the statement made me expect a "positive" result. However, 5.6 % is not exactly a promising result, despite the identification rate being indeed higher than that from aerobic bottles (0 %).

L.42: "except for the anaerobic bottle": this result was already stated in L.31.

L.67: Delete the comma in "BSI,".

LI.90-93: "recently introduced Rapid Sepsityper (RE) protocol": introduced by who? Experimental protocol or commercially available kit (FDA? IVDR?)?

L.94: Is "Bruker Daltonic GmbH" the same company as the aforementioned "Bruker Daltonics, Bremen, Germany"?

L.117: The clinical strains should be characterized: were they bloodstream isolates?

LI.227-229: "significantly improved [...] 98.2 % vs. 100 %" - please check this statement. Is it possible that the significant improvement occurred in comparison to the groups with no lysis buffer, with no reagent modification, or the group with the replaced washing buffer? In the current status, it sounds like you compared the difference between 98.2 % and 100 % ("vs.").

L.230: Result for the group with no lysis buffer?

LI.240-242:

LI.253-255: Did the BC bottles become positive after 16 h or did they never become positive? Was it the same clinical strain in all four bottles?

L.279: This is the first time that supplementary figure 1 is mentioned. Since it is a result, it should appear in the results section. Since it is even an interesting result, it should be considered to transfer it from the supplementary part to the regular figures.

LI.305-309: The alternative protocols should be discussed in more detail: Why do they work better? What are their disadvantages? Are they a better alternative to the very simple "life hack" to just replace lysis and washing buffer with PBS upon the finding of streptococci suspicious for pneumococci in the Gram stain?

Figure 1: It seems that the picture of the Bruker devices features the smart system instead of the sirius system.

Figure 2: According to the scale bars, the lower picture is magnified by the factor of 2.5 compared to the upper picture ("5.00um" vs. "2.00um"). This fits well for panels B and C. However, the upper picture of panel A seems questionable: there are no coccal and particularly no diplococcal structures visible in in this picture. It looks like crystal-like structures. With a higher magnification (only factor 2.5!), the crystals suddenly turn to clearly recognizable (pneumo)cocci? Please check whether this is the correct picture. By the way, I really appreciate that the authors present such data, since "just looking at the bacteria" is often neglected in comparable studies - thanks!

Dr. Tulip Jhaveri

Editor

Microbiology Spectrum

Dear Dr. Jhaveri,

Please find enclosed the revised manuscript entitled “*Experimental study on factors influencing low identification rates and spectral quality of Streptococcus pneumoniae using the Sepsityper Kit.*”

We are very grateful for your continued consideration of our manuscript. We would also like to thank Reviewer #1 for their critical comments and constructive suggestions, which have greatly improved the quality of our work. Our responses to the reviewer’s comments are provided below in a point-by-point format. For your convenience, all changes in the revised manuscript have been marked using the “Track Changes” function in Microsoft Word.

We hope that the revised version will be acceptable for publication in *Microbiology Spectrum*.

Yours sincerely,

Hyun-Seung Lee

on behalf of all authors

Hyun-Seung Lee

Department of Laboratory Medicine, School of Medicine, Wonkwang University, Iksan,
Republic of Korea

E-mail: ctmasaru@gmail.com

Reviewer #1

Since I was already involved in a previous review process (reviewer #3), my major comments have all been answered by the authors - thank you. Despite some minor issues, which were not considered and which are not listed in the "response to reviewers" file (?), this manuscript is suitable for publication in Microbiology Spectrum.

Please find below the minor comments as stated in my original review. Please note that the line numbers might have changed due to the newer modifications of the manuscript.

► Authors' response: We sincerely thank you for your encouraging comments and constructive suggestions. We hope that the revisions made to the manuscript adequately address your concerns.

1. L1.31-32: Concerning the information about higher identification rates from anaerobic BC bottles in the abstract, I would like to propose to either delete this statement here or to add the information about the identification rate. At least for me, it was misleading, since the statement made me expect a "positive" result. However, 5.6 % is not exactly a promising result, despite the identification rate being indeed higher than that from aerobic bottles (0 %).

► Authors' response: Thank you for your suggestion. In response, we have revised the sentence to include the identification rates, as follows:

“The results showed that anaerobic bottles had a higher identification rate than aerobic bottles (5.6% vs. 0%).”

2. L.42: "except for the anaerobic bottle": this result was already stated in L.31.

► Authors' response: Thank you for your comment. As suggested, we have removed the redundant phrase.

3. L.67: Delete the comma in "BSI,".

► Authors' response: Following your suggestion, we have removed the comma after “BSI”.

4. L1.90-93: "recently introduced Rapid Sepsityper (RE) protocol": introduced by who? Experimental protocol or commercially available kit (FDA? IVDR?)?

► Authors' response: Thank you for your comment. We have clarified that the Rapid Sepsityper protocol was introduced by the manufacturer, as follows.

“To improve efficiency, Bruker Daltonics recently introduced the Rapid Sepsityper (RS) protocol, which eliminates the protein extraction step. This protocol reduces hands-on time

from 20 to 30 min to less than 10 min and enables identification results within 15 to 20 min of a positive blood culture alert.”

5. L.94: Is "Bruker Daltonic GmbH" the same company as the aforementioned "Bruker Daltonics, Bremen, Germany"?

► Authors' response: Bruker Daltonic GmbH is the same company as the previously mentioned "Bruker Daltonics." To avoid confusion, we have changed the name "Bruker Daltonic GmbH" to "Bruker Daltonics.”

6. L.117: The clinical strains should be characterized: were they bloodstream isolates?

► Authors' response: Thank you for the important comment. We have clarified the description of the clinical strains as “five clinical strains of *S. pneumoniae* isolated from patients with BSI.”

7. L1.227-229: "significantly improved [...] 98.2 % vs. 100 %" - please check this statement.

Is it possible that the significant improvement occurred in comparison to the groups with no lysis buffer, with no reagent modification, or the group with the replaced washing buffer? In the current status, it sounds like you compared the difference between 98.2 % and 100 % ("vs.").

L.230: Result for the group with no lysis buffer?

► Authors' response: Thank you for pointing this out. We recognize that the original sentences may have caused confusion and have revised the sentences accordingly to enhance clarity.

“Replacing the lysis buffer (98.2%, 53/54) or both the lysis and washing buffers (100%, 54/54) in the Sepsityper with PBS significantly improved species-level identification rates for *S. pneumoniae*. These rates were markedly higher than those observed in the group with no reagent modification (48.2%, 26/54) and the group in which only the washing buffer was replaced with PBS (42.6%, 23/54) (Table 1).”

8. L1.253-255: Did the BC bottles become positive after 16 h or did they never become positive? Was it the same clinical strain in all four bottles?

► Authors' response: We acknowledge that the original sentences were unclear and potentially confusing. Accordingly, we have revised them as follows for improved clarity.

“In the validation study, 16 of the 20 blood culture bottles spiked with five clinical strains of *S. pneumoniae* produced positive alarms within 16 h. The remaining four bottles—one BACT/ALERT anaerobic bottle and three BD BACTEC Plus Aerobic/F bottles—produced positive alarms after 16 h and were excluded from the analysis due to suspected technical errors. Clinical strains spiked in the excluded four bottles were as follows: serotype 3 ($n = 2$), serotype 35B ($n = 1$), and serotype 15C ($n = 1$).”

9. L279: This is the first time that supplementary figure 1 is mentioned. Since it is a result, it should appear in the results section. Since it is even an interesting result, it should be considered to transfer it from the supplementary part to the regular figures.

► Authors' response: Thank you for the constructive suggestion. We assumed you were referring to Supplementary Figure 2 and accordingly have moved it to Figure 3. We have also added the corresponding finding to the Results section as follows.

“Mass spectra randomly selected from the validation study were compared to a mass spectrum of *S. pneumoniae* grown on a BAP. Despite correct species-level identification, the number of mass peaks obtained using the Sepsityper—regardless of the extraction method—was markedly lower than that of the BAP-grown isolate (Figure 3).”

10. L1.305-309: The alternative protocols should be discussed in more detail: Why do they work better? What are their disadvantages? Are they a better alternative to the very simple "life hack" to just replace lysis and washing buffer with PBS upon the finding of streptococci suspicious for pneumococci in the Gram stain?

► Authors' response: Thank you for your comments and suggestions. We have accordingly expanded the discussion of alternative protocols as follows.

“Given these findings, alternative approaches may be more reliable for identifying *S. pneumoniae* in positive blood culture broth (33, 35). The SST tube method, which removes RBCs without the use of a lysis buffer, achieved a 100% identification rate (8/8) (33). Similarly, saponin-based enrichment, which uses saponin as an RBC lysis solution, achieved

an 80% identification rate (4/5) (35). Molecular approaches or modified bile solubility test can also be performed directly on positive blood culture broth (48, 55). However, despite their superior performance in *S. pneumoniae* identification, these alternative methods are labor-intensive or costly, limiting their widespread adoption in routine diagnostics. Moreover, simply omitting the lysis buffer from the Sepsityper Kit may not be an effective strategy for improving *S. pneumoniae* identification rates. Hariu et al. reported that *S. pneumoniae* was not identified when the lysis buffer was omitted, in contrast to *Escherichia coli*, *Klebsiella pneumoniae*, and *Pseudomonas aeruginosa*, which were successfully identified without it (38). One should bear in mind that, despite the widespread use of rapid diagnostics, Gram staining remains an essential test for evaluating positive blood cultures, and subculturing remains the gold standard for accurate identification—particularly for *S. pneumoniae*—albeit at the cost of additional time.”

11. Figure 1: It seems that the picture of the Bruker devices features the smart system instead of the sirius system.

►Authors’ response: In response to the previous suggestion from another reviewer, we have removed Figure 1, which contained images of the Bruker devices, from the manuscript.

12. Figure 2: According to the scale bars, the lower picture is magnified by the factor of 2.5 compared to the upper picture ("5.00um" vs. "2.00um"). This fits well for panels B and C. However, the upper picture of panel A seems questionable: there are no coccal and particularly no diplococcal structures visible in in this picture. It looks like crystal-like

structures. With a higher magnification (only factor 2.5!), the crystals suddenly turn to clearly recognizable (pneumo)cocci? Please check whether this is the correct picture. By the way, I really appreciate that the authors present such data, since "just looking at the bacteria" is often neglected in comparable studies - thanks!

► Authors' response: Thank you for your comment. While you referred to Figure 2, we believed you were referring to Supplementary Figure 1. The image displaying crystal-like structures appears to be a technical artifact and could potentially confuse readers. Therefore, we have removed the upper images from Supplementary Figure 1, as the lower images alone adequately illustrate our findings.

Re: Spectrum01084-25R1 (Experimental study on factors influencing low identification rates and spectral quality of *Streptococcus pneumoniae* using the Sepsityper Kit)

Dear Dr. Hyun-Seung Lee:

Your manuscript has been accepted, and I am forwarding it to the ASM production staff for publication. Your paper will first be checked to make sure all elements meet the technical requirements. ASM staff will contact you if anything needs to be revised before copyediting and production can begin. Otherwise, you will be notified when your proofs are ready to be viewed.

Sincerely,
Tulip Jhaveri
Editor
Microbiology Spectrum